# The Human Ntn-Hydrolase Superfamily: Structure, Functions and Perspectives

**DOI:** 10.3390/cells11101592

**Published:** 2022-05-10

**Authors:** Arne Linhorst, Torben Lübke

**Affiliations:** Department of Chemistry, Biochemistry I & III, Bielefeld University, 33615 Bielefeld, Germany; arne-linhorst@web.de

**Keywords:** N-terminal nucleophile (Ntn)-hydrolases, Ntn-fold, auto-activation, amide bond cleavage, asparaginases, proteasome subunits, penicillin acylases, γ-glutamyl transpeptidases

## Abstract

N-terminal nucleophile (Ntn)-hydrolases catalyze the cleavage of amide bonds in a variety of macromolecules, including the peptide bond in proteins, the amide bond in N-linked protein glycosylation, and the amide bond linking a fatty acid to sphingosine in complex sphingolipids. Ntn-hydrolases are all sharing two common hallmarks: Firstly, the enzymes are synthesized as inactive precursors that undergo auto-proteolytic self-activation, which, as a consequence, reveals the active site nucleophile at the newly formed N-terminus. Secondly, all Ntn-hydrolases share a structural consistent αββα-fold, notwithstanding the total lack of amino acid sequence homology. In humans, five subclasses of the Ntn-superfamily have been identified so far, comprising relevant members such as the catalytic active subunits of the proteasome or a number of lysosomal hydrolases, which are often associated with lysosomal storage diseases. This review gives an updated overview on the structural, functional, and (patho-)physiological characteristics of human Ntn-hydrolases, in particular.

## 1. Introduction

Amide bonds and amide-related bonds are characterized by a high resistance towards hydrolysis under physiological conditions, which is attributed to the partial double bond character of the mesomerically stabilized CO-N bond [1]. As a consequence, evolution favored amide bonds in many biomolecules such as peptides, proteins, DNA, RNA, or complex lipids (e.g., ceramides) [2]. Here, we review a particular group of amide bond-cleaving enzymes, called N-terminal nucleophile (Ntn)-hydrolases, which specifically hydrolyzes amide bonds, by taking use of a characteristic protein fold as well as by undergoing a family-specific way of auto-activation [3]. Due to the growing complexity and functional diversity of this family, we primarily focus on eukaryotic representatives that, according to their hydrolytic reaction, can be classified into EC 3.5.1 (hydrolases acting on carbon-nitrogen bonds in linear amides, other than peptide bonds) of the BRENDA enzyme database. However, some representatives of these amidases are also, more specifically, grouped in EC 3.4.19 (hydrolases acting on isopeptide bonds/omega-peptidases) and EC 3.4.25 (threonine endopeptidases), respectively, with regard to their known or assumed function.

## 2. The Ntn-Hydrolase Superfamily

By means of crystal structure analyses derived from prokaryotic and eukaryotic amidohydrolases, including glutamine-phosphoribosylpyrophosphate-amidotransferase (GAT, *B. subtilis*) [4], penicillin G-acylase (PGA, *E. coli* and *P. rettgeri*) [5], and the 20S proteasome (Pr, *T. acidophilum*) [6], a characteristic αββα-core fold was independently described, sharing substantial structural homology rather than sequence homology. Brannigan et al. initially summarized these findings and classified the proteins as a novel structural superfamily, which was termed the N-terminal nucleophile (Ntn)-hydrolase superfamily, as all identified members share a characteristic catalytic serine, threonine, or cysteine nucleophile at their amino termini, upon autoactivation within the characteristic αββα-fold. It was also concluded that this particular fold constitutes a catalytic framework, in order to enable the autocatalytic activation step leading to the N-terminal nucleophile, which, consequently, also forms the catalytic active site of this enzyme family [3]. Due to the presence of these common Ntn-specific hallmarks, the human lysosomal aspartylglucosaminidase (AGA) [7], the penicillin V acylase (PVA) from *B. sphaericus* [8] and the γ-glutamyl transpeptidase (GGT) from *E. coli* [9] were also, subsequently, added as subfamilies of this superfamily (see Table 1).

Lately, the SCOPe (Structural Classification Of Proteins—extended) database listed a seventh subfamily with an N-terminal threonine nucleophile (d.153.1.7), which solely comprises the so-far uncharacterized bacterial protein SPO2555 [10]. Anyways, SPO2555 does not necessarily define a discrete subfamily, as the domain structure strongly resembles that observed in the proteasome subfamily (PDB 2imh); furthermore, no peer-reviewed data are available.

Meanwhile, a still-growing number of family members have been added to the different subfamilies, including well-known and newly identified proteins, from bacteria to mammals, such as the cephalosporin acylase (CA, *P. diminuta*) [11], acid ceramidase (ASAH1) [12] or the phospholipase B-like proteins 1 and 2 (PLBD1 and PLBD2, respectively) [13,14].

## 3. Ntn-Hydrolase Consensus Fold

The crystal structures, as well as functional studies of the originally identified Ntn-hydrolases (GAT, PGA, Pr, AGA), helped to describe the common consensus αββα-fold that is conserved as unexceptional in all Ntn-hydrolase family members [3]. The four-layered configuration of secondary structural elements includes two α-helices in the αI-layer, five and four antiparallel beta-sheets in the βI- and βII-layer, respectively, and two further α-helices in the αII-layer [15]. While these secondary structure elements show conserved spatial configuration, their interconnections and the position of additional α-helices and β-sheets differ in a distinct subfamily-specific manner [16] (Figure 1).

All known Ntn-hydrolases are subjected to an autocatalytic endopeptidic cleavage at the eponymous nucleophilic residue, which is either a threonine, serine, or cysteine, resulting in the generation of an N-terminal α-fragment and a C-terminal β-fragment. This cleavage releases the amino group of the threonine, serine, or cysteine from the peptide bond and, thus, leaves this residue at the N-terminus of the newly created β-fragment. As a consequence, the N-terminal nucleophile is involved in two important functions: (1) initiation of autocleavage and (2) catalysis of the appropriate substrate [3,16]. The corresponding smaller α-fragments of the Ntn-hydrolases, which are variable in size and in physiological relevance for enzymatic activity of the matured protein, are almost always retain associated, either non-covalently or covalently by disulfide bridges, with the larger β-fragments [3,16]. Moreover, quaternary structures such as heterotetramerization (αβ)_2_ (AGA, Plbd2) or even high-molecular oligomeric complexes (Pr) are formed, which might be critical for overall hydrolytic activity [6].

At the newly created C-terminus of the smaller α-fragment, a variable number of amino acids are removed by restricted degradation, which—with respect to the substrate—often leads to a facilitated access towards the catalytic center [17,18,19]. Such a concept, of a disposable peptide-linker that covers the catalytic site of inactive precursors, is widely seen, e.g., in lysosomal enzymes, thus preventing cellular damage during lysosomal transport passing the ER and Golgi apparatus [20]. Upon autocatalytic cleavage, these linker peptides become structurally flexible and might undergo further, despite restricted degradation to facilitate substrate access to the catalytic center [21]. In the PGA from *E. coli* and other members of this subfamily, the removal of the linker is essential for enzyme activity [22], while, in contrast, in human AGA the removal of a rather small linker fragment was shown after autocatalytic cleavage [23], which has no obvious influence on catalytic activity [24].

## 4. Active Site Residues

In contrast to the commonly known catalytic triads of the serine proteases, the Ntn-hydrolase active site was considered to bear just a single active site residue [25], in which the free α-amino group of the nucleophile functions as a general base [5]. A more complex view, however, describes several critical amino acids in the active site that exert a supporting effect, by forming an oxyanion hole or stabilizing the nucleophile, the general base, and the substrate [26]. In the AGA, an additional stabilizing effect of the substrates’ α-amino group towards the nucleophilic oxygen of the catalytic threonine was proposed, in order to reduce the pKa value of the threonine [27] and to allow rather high enzymatic activity at low lysosomal pH values [28]. The presence of an active-site cysteine constitutes another optimization towards acidic environments, as the lower pKa value of the cysteine thiol group compared to hydroxy groups in serine or threonine simplifies the deprotonation. Consequently, the conjugated bile acid hydrolase (CBAH) from the PVA family come along with a pKa of the Ntn-thiol group as low as 4 and a pKa of the α-amino function of around 7.5 [29], resulting in a pH-optimum between 5.8 to 6.4, in which the active site cysteine exists in a zwitterionic form [30].

As the gamma (γ) oxygens of serine and threonine are chemically similar, the hydrophobic γ-methyl group of the threonine is the major difference between these two nucleophiles. In the active site of classical serine proteases, the methyl function of a threonine (instead of a serine) would either block the interaction of the nucleophile and the general base, or would clash with the oxyanion generated during catalysis [31]. In contrast, the nucleophile in Ntn-hydrolases allows the positioning of the α-amino function in a curved manner, thereby avoiding the steric hindrance of the threonine γ-methyl group with the general base. Beyond that, the hydrophobic interaction with the γ-methyl group allows the precise orientation of the hydroxyl functionality towards the catalytic center, assisting the autocatalytic cleavage [32] and allowing higher catalytic efficiency [33]. Consistently, upon mutation of the catalytic threonine to serine, the activity of the proteasome (Pr) from T. acidophilum [34] as well as the E. coli ASNase3 drops by an order of magnitude [35].

## 5. Further Superfamilies with Ntn-Hydrolase-like Folds

It was suggested to introduce another Ntn-hydrolase subfamily of β-aminopeptidases, as reflected by the L-aminopeptidase D-Ala-esterase/amidase (DmpA) as a prototype [36], which structurally and functionally resembles the Ntn-hydrolases (DmpA, *O. anthropi*) [37]. However, an in-depth analysis of its evolutionary origin instead assigned it to a new superfamily, with convergent evolution to the Ntn-hydrolases [38]. Moreover, the archaeal IMP cyclohydrolase (PurO, [39]) exhibits a Ntn-hydrolase-like fold that might have evolved from the same precursor but developed towards a different enzymatic function, defining another superfamily [40].

## 6. Ntn-Hydrolases in Human

The Ntn-hydrolase superfamily includes a large number of prokaryotic, archaeal, and eukaryotic proteins (>130 identifiers in MEROPS Database Clan PB [41]). Assignment is based on the identification of the characteristic Ntn-hydrolase fold in the crystal structure or from sequence homology to known Ntn-hydrolases. The number of Ntn-hydrolases in humans, however, is quite manageable, but it can be confusing due to varying synonyms, as names often changed when new functional insights are acquired. Human Ntn-hydrolases carry either cysteine or threonine nucleophiles, as reflected by the letter-code in their MEROPS family identifier (e.g., C44 or T1, see Table 2). The bacterial serine-nucleophile subfamily PGA is not found in humans.

### 6.1. Family of Class II Glutamine Amidotransferases (GAT)

Glutamine amidotransferases (GAT) releases ammonia from glutamine (omega-peptidase-like activity) that is utilized in a secondary subunit for transferase (EC2) and ligase (EC6) reactions on a variety of substrates [42]. While class I amidotransferases (trpG-type/G-type) act via a Cys–His–Glu catalytic triad, only class II amidotransferases (purF-type/F-type) belong to the Ntn-hydrolase family [43]. Due to their catalytic cysteine nucleophile, these enzymes are grouped in MEROPS Family C44, with the N-terminal cysteine being preceded by a few amino acids that are autocatalytically cleaved. The ammonia released in the class II GAT domain is transferred efficiently to the active site of the second domain by an intramolecular channel [44], where the products asparagine (ASNS, ASNSD1), phosphoribosylamine (GPAT), or glucosamine-6-phosphate (GFAT1, GFAT2) are generated.

In the glutamine-dependent asparagine synthetase (ASNS, Figure 2), the C-terminal domain transfers the released ammonia to aspartate in an ATP-dependent manner to generate asparagine [45]. With the ASNS being the only enzyme performing asparagine neosynthesis in humans, deficiency leads to a severe disorder called asparagine synthetase deficiency [46]. The disease is characterized by severe neurological signs, impaired psychomotor development, and mortality at an early age; it can only, to a low extent, be treated with dietary asparagine [47]. Elevated ASNS expression, on the other hand, is related to resistance against asparaginase therapy in childhood ALL, as the usually low capacity for asparagine synthesis in leukemia cells leads to starvation upon asparagine depletion from the plasma [48] (see section below on Family of Asparaginases). With the asparagine-synthetase-domain-containing protein 1 (ASNSD1), another potential GAT class II enzyme was predicted by similarity.

The glutamine-phosphoribosylpyrophosphate amidotransferase (GPAT, also named PPAT) contains a class II GAT domain and a C-terminal domain of the type I phosphoribosyltransferases (PRTase) [49]. As the first step in de novo purine biosynthesis, the released ammonia is used to substitute a pyrophosphate and generate phosphoribosylamine. This rate-limiting step is tightly regulated via feedback inhibition and conformational changes [50]. On an intermolecular basis, GPAT is part of a larger complex called the purinosome, combining the six enzymes of the purine biosynthesis [51]. GPAT associated diseases are not described in humans, and GPAT-deficient-mouse models are not viable [52]. Even though salvage pathway purine syntheses might still be functional, defects in other purine de novo metabolic enzymes of the purinosome lead to severe diseases as well [53].

Glutamine-fructose-6-phosphate amidotransferase (GFAT) is the enzyme performing the syntheses of glucosamine-6-phosphate (GlcN-6-P) from its substrate fructose-6-phosphate (Fru-6-P), using glutamine as the amino group donor [54]. The ammonia released from the class II GAT domain is channeled to the sugar isomerase domain, as shown for the bacterial homologue GlmS [44]. The two isoforms, GFAT1 and GFAT2, are expressed in a tissue-specific manner [55] and catalyze the first and rate-limiting step in the hexosamine biosynthesis pathway [56]. At least, the GFAT1 enzymatic activity is tightly regulated by the nutrition sensor mTORC2 [57]. GFAT1 deficiency leads to failure in hexosamine metabolism and becomes most evident in the development of a congenital myasthenic syndrome (CMS), resulting from the lack of essential glycosylated proteins in the nerve-muscle synapse [58]. GFAT2 upregulation, on the other hand, is associated with increased susceptibility for type 2 diabetes mellitus [59].

In humans, the majority of the amidotransferases using glutamine as a substrate are of either the GAT subfamilies, with the exception of the mitochondrial Gln-tRNA-amidotransferase (GatA), or the glutamine-dependent NAD^(+)^-synthetase (NADE). GatA is assigned to the amidase signature superfamily [60] and, structurally, clearly differs from the Ntn-hydrolases, as shown for a prokaryotic homologue [61]. The glutaminase domain in NADE shows an αββα-fold tertiary structure such as the Ntn-hydrolases [62], but does not share other distinct Ntn-typical features such as antiparallel β-sheets. NADE was assigned to the nitrilase superfamily of carbon-nitrogen hydrolases [63].

### 6.2. PVA-Subfamily of Lysosomal Hydrolases

The enzymes from MEROPS families C89, C95, and C69 account for the diverse, and only partially characterized, subfamily of cysteine-nucleophile Ntn-hydrolases. The eponym Penicillin V Acylase (PVA) represents the exemplary structure of this subfamily, but is not found in humans. The catalytic active cysteine nucleophile of human PVA-subfamily members is found to be rather central within the primary sequence, resulting in similarly sized α- and a β-subunits after cleavage. As these subunits stay attached to each other in the mature enzyme, the C-terminal part of the α-subunit has to dislocate from the active cleft to allow substrate accessibility. While the family C89 members (AC, NAAA) hydrolyze fatty acid-amide bonds, the substrates of family C95 (PLBD1/2) and C69 (SCRN1-3) members remain elusive.

The lysosomal acid ceramidase (AC, ASAH1, Figure 2) is an Ntn-hydrolase deacetylating ceramide to generate sphingosine and free fatty acid, so that AC deficiency leads to the lysosomal storage disorder Farber disease (FD; lipogranulomatosis) [64] and spinal muscular atrophy with progressive myoclonic epilepsy (SMA-PME), where the latter is a rare condition with only around 200 cases reported [65]. By the cleavage of ceramides into sphingosines and fatty acids, AC participate in the regulation of important bioactive lipids, involved in cell cycle control, apoptosis, or inflammation [66]. Furthermore, in different glycosphingolipid storage disorders, AC is involved in the transformation of pathologic storage material [67]. The active enzyme is generated by autocatalytic cleavage under acidic conditions at the cysteine in position 143 of the 53 kDa precursor, leading to the mature heterodimeric protein, composed of an unglycosylated 13-kDa α- and a glycosylated β-subunit with an apparent molecular weight of about 40 kDa [68,69], which are covalently attached to each other by disulfide bonds [70]. For the extraction of the hydrophobic ceramide substrate from membrane surfaces, AC is dependent on the cofactor protein saposin D that transfers the lipid substrate towards the hydrophobic active cavity [71]. Upon deficiency of the saposin D cofactor, acid ceramidase activity is largely impaired [72]. The functionally homologous alkaline ceramidases (ACER1, ACER2, ACER3) and the neutral ceramidase (ASAH2), on the other hand, do not show Ntn-hydrolase characteristics [73,74].

Being closely related to acid ceramidase (33% identity), the lysosomal N-acyl-ethanolamine-acid amidase (NAAA) also releases free fatty acid as a product, but acts on the fatty-acid-amides of the N-acyl-ethanolamine (NAE) group of lipids. As NAEs constitute a group of highly potent bioactive lipids, with anti-inflammatory or anti-anaphylactic properties [75], it is consistent that NAAA-deficient mice showed an impaired response to triggers of inflammatory skin diseases [76]. Accumulation of lysosomal storage material, as described for deficiency models of acid ceramidase, was not reported in NAAA-deficient mice, possibly due to a compensatory effect of the functionally related fatty acid amide hydrolase (FAAH) [77].

Maturation of the NAAA precursor (glycosylated ~49-kDa form) implies limited self-proteolysis by cysteine 126, whose thiol side chain arises to the catalytic nucleophile. The resulting catalytically active heterodimer is preferentially formed in the acidic milieu of the lysosomal compartment and, finally, consists of a smaller α-chain (~19 kDa) and a 30-kDa β-subunit [78]. A key function for activity was attributed to two hydrophobic helices, namely α3 and α6, which were shown to interact with lipid membranes to induce a conformational change that allows extraction of the NAE-substrate, from the membrane to the active site. This interfacial activation mechanism is thought to substitute for the role of the lipid activator protein saposin D in AC [79].

With phospholipase B domain containing 2 (PLBD2), another lysosomal protein was assigned to the Ntn-hydrolase superfamily, after the crystal structure of the murine protein was solved (see Figure 3). Maturation towards the putative active form requires multiple steps of limited proteolysis and removal of a linker fragment between the α- and β-subunit (Figure 4) [17]. PLBD2 was categorized into the PVA subfamily from its homology in active site residues, but, among the Ntn-hydrolases, only shares a notable sequence identity with the homologous protein PLBD1 [80].

While deficiency of the PLBD2-orthologous protein p67 in the protozoa *Trypanosoma brucei* leads to severe lysosomal alterations and premature death [81], the suspected phenotype of a lysosome-associated disease was not observed in PLBD2-deficient mice (Linhorst, Lübke, unpublished data). The specific substrate remains enigmatic, but is likely to be found among fatty acid amide-linked lipids, as described for the other human PVA family members AC and NAAA. As both PLBD1 and PLBD2 are highly expressed in immune cells, namely neutrophils [82], monocytes [83] and bone-marrow-derived macrophages (Linhorst, Lübke, unpublished data), a function in innate immunity is conceivable.

With the growing therapeutic relevance of biopharmaceutical proteins from recombinant expression systems, PLBD2 has received much attention as a common host cell protein (HCP) contaminant in drug product formulations [84,85,86]. Copurification was shown for several CHO-cell-derived monoclonal antibody formulations, where HCP impurity may result in anti-PLBD2 specific immune response upon administration in patients. Occurrence of CHO-PLBD2-specific antibodies was comprehensively described for an anti-IL13 monoclonal antibody [87] and may be avoided by the application of PLBD2-deficient host cells [88].

MEROPS family C69 contains the enigmatic secernin proteins (SCRN1, SCRN2, SCRN3). Without crystal structure data available, the assignment to the PVA subfamily of the Ntn-hydrolases is based on the homology of conserved amino acids surrounding the putative active site [89], which was recently supported by the computational prediction of the typical Ntn-fold in those enzymes [90].

While the precursor proteins of SCRN2 and SCRN3 are cleaved upon overexpression, most likely by their autocatalytic activity, the SCRN1 precursor remains unchanged [91]. This finding corresponds to the presence of a serine, instead of the PVA-typic cysteine, at the active site [89]. In PLBD2, a mutation of the active-site cysteine towards a serine, likewise, leads to a loss of autocatalytic cleavage (Linhorst, Lübke, unpublished data). The cytosolic SCRN1 was shown to be involved in endoplasmic reticulum (ER) remodeling at presynaptic sites in neurons and exocytosis in mast cells [91,92], most likely using a FFAT (Phe-Phe-acid-tract)-motif that was initially found in ER-associated cytosolic lipid binding proteins [93]. The exclusive presence of the FFAT-motif in SCRN1, together with the lack of autocatalytic cleavage, points towards an entirely different, non-hydrolytic function of SCRN1 compared to SCRN2 and SCRN3, which possibly could have evolved from a lipid-amide cleaving activity of the PVA-hydrolases.

As mentioned before, the cysteine residue in the active site of Ntn-hydrolases can be considered an optimization towards acidic environments. A hydrolytic activity of SCRN2 and SCRN3 in the endo-lysosomal compartment, as shown for AC and NAAA and expected for PLBD1 and PLBD2, is, therefore, highly probable. Even though a homology towards the U34-dipeptidase family may suggest peptidase-like activity [89], neither the physiological localization nor the true function of SCRN2 and SCRN3 has been revealed experimentally.

### 6.3. Family of Proteasome Subunits (Pr)

The core 20S proteasome (20S CP; ~750 kDa) is a multi-subunit catalytic protease complex, for the non-lysosomal protein degradation within the cytosol of eukaryotes. However, comparable multicatalytic proteinases are also found in archaebacteria (1PMA) [6] and, even, in some prokaryotes. In electron microscopy, the eukaryotic (as well as the archae) 20S CP offers itself as a cylinder-shaped structure that is 15 nm in length and 11 nm in diameter, composed of 28 subunits (two copies each of 14 subunits), which are axially stacked in four homo-heptameric rings [94,95]. In eukaryotes, a set of seven related α-subunits (α1-α7), which are all proteolytically inactive, controls access to the catalytic core, which, for its part, is formed by seven β-subunits (β1-β7), of which three are proteolytically active. The assembly of the resulting α7β7β7α7 architecture takes hours and, additionally, needs maturation factors [96]. Within the complex, a central channel including two smaller chambers and one larger central chamber is formed, in which degradation of ubiquitinated protein substrates is carried out [6,94,97]. The three proteolytically active β-subunits β1 (Y/Pre3), β2 (Z/Pup1), and β5 (X/Pre2) in human/yeast, respectively, of the 20S CP, possess the characteristic Ntn αββα-fold and mature autocatalytically to the active subunits, by revealing a nucleophilic threonine at their newly created N-termini [94,97]. Also worth noting, the inactive α- and β-subunits share the Ntn-fold but, most likely, just serve as a scaffold for the active subunits. After superimposing the bovine 20S CP into the yeast proteasome crystal structure, it was also proposed that the β7-subunit might mediate Ntn-based proteolytic activity, although this additional activity was not confirmed later on [95]. In vertebrates, interferon-γ induces the synthesis of alternative Ntn-related β-subunits β1i (*LMP2*), β2i (*MECL-1*) and β5i (*LMP7*), hence, resulting in the formation of the so-called immuno-proteasome, which generates peptides that can be further trimmed to efficiently bind into the groove of MHC class I molecules [98]. Finally, a β5t-subunit exclusively expressed in the thymus generates unique peptides that are relevant for T-cell selection [99].

### 6.4. Family of Asparaginases (AGA)

Asparaginases across species can be classified according to their similarity to prototypic enzymes from *E. coli* cytosol (class 1/type I), *E. coli* secretome (class 1/type II), plant (class 2/type III), or *Rhizobium etli* (class 3/type IV and V) [100]. The type III asparaginases (class 2) are Ntn-hydrolases that are found in the MEROPS T2 family (threonine-type isoaspartyl dipeptidases). In humans, these include the lysosomal aspartyl glucosaminidase (AGA), an isoaspartylpeptidase (ASRGL1), and an aspartyl endopeptidase (TASP1). Moreover, members of the asparaginase type III family are also found in plants and prokaryotes, such as EcAIII in *E. coli*, which functions in the degradation of isoaspartyl-containing di-peptides but, also, might be able to release aspartate and ammonia from asparagine [101].

A similar dual functionality covering asparaginase and isoaspartyl-peptidase activity was demonstrated for the human asparaginase ASRGL1 (syn. hASNase3, ALP, CRASH, EC3.4.19), which owes autoproteolytic activation to threonine in position 168 [102,103]. In contrast to many other Ntn-hydrolases, human ASRGL1 undergo a very slow autocleavage in vitro, which became obvious in crystallization experiments, where under neutral pH the majority of the protein remains as uncleaved protein, forming, primarily, a homodimer of two precursors [102]. A recent study, moreover, suggests trimerization as the key to complete processing [104]. The catalytic site needed for asparagine hydrolysis includes, besides Thr168, two further threonines (Thr186, Thr219), as well as Arg196 and Asp199, as concluded from comparisons with known structures from EcAIII and hAGA [7,105]. Presumed activity of ASRGL1 involves degradation of potentially toxic β-aspartyl peptides [103], as was initially proposed for all plant-type asparaginases [106].

The human lysosomal aspartylglucosaminidase (AGA; EC 3.5.1.26, Figure 2) is another homologues protein of the *E. coli* EcAIII asparaginase. AGA is involved in the degradation of N-linked glycoproteins, more precisely in the hydrolysis of the N-glycosidic linkage between the N-actelyglucosamine at the reducing end of the oligosaccharide and an asparaginyl-residue within the glycoprotein backbone. During the reaction cycle, aspartic acid is left behind at the protein moiety, while the 1-aminoglycan intermediate product is further hydrolyzed into ammonia and the appropriate oligosaccharide [107]. The AGA crystal structures served as one of the Ntn’s “founding members” [7,108], thus, all Ntn-related aspects are fulfilled. Both the precursor and the mature form exhibit the characteristic αββα-fold, where Thr183 takes over dual responsibility in terms of autoprocessing towards the active mature form as well as for catalytic activity as an N-terminal nucleophile in the activated lysosomal protein. The autocleavage of the precursor forms results in the occurrence of α-chain/β-chain heterodimers, which, in terms of a quaternary structure, form a hetero-tetrameric AGA molecule [7]. The oligomerization, mainly the dimerization of Ntn-hydrolases, however, seems to be ascribed to a stabilization effect rather than to a cooperative catalytic purpose.

AGA is of particular medical interest as its enzymatic deficiency, aspartylglucosaminuria (AGU), leads to the lysosomal accumulation of undegraded glyco-asparagines, in consequence of which intellectual disabilities, seizures, and ataxia occur, due to the particularly loss of nerve cells [109]. To date, no curative therapy exists for AGU as a classical enzyme replacement therapy, which is well established for around 10 lysosomal storage diseases; thus, it is hardly feasible in diseases with neuronal involvement, due to bloodbrain barrier restrictions [110]. Currently, therapeutic approaches are underway, including the use of pharmacological chaperone to stabilize mutant forms of AGA as well as adeno-(associated) gene transfer strategies [111,112].

The third known member of the asparaginase-subfamily shows aspartyl endopeptidase activity (official gene name TASP1; EC3.4.25) and was, initially, identified as the protease responsible for the site-specific proteolysis of the HOX-gene regulator protein MLL (Mixed-lineage Leukemia) [113]. The cleavage site was found to comprise an aspartic acid in P1 and a glycine in P1′ position [114]. However, with the other two family members acting in omega-peptidase reactions on asparagine-side-chain residues and with the susceptibility of the Asp-Gly-dipeptide for aspartyl isomerization [115], it is tempting to speculate on an isoaspartyl formation as the origin of endoproteolytic TASP1 activity. TASP1 is synthesized as a 50kDa inactive precursor and undergoes autocleavage that leads to a 28-kDa-N-terminal (α)-fragment and a C-terminal 28-kDa-β-fragment, which owns the nucleophilic Thr234 at its outmost N-terminus and, hence, proteolytic activity against its most prominent substrate MLL is dependent on Thr234 [113]. Recently, loss-of-function variants of TASP1 were associated with a severe intellectual disability syndrome that resembles diseases, which are caused by defective histone-modifying methyltransferases [116], reflecting the general impact of TASP1 function on the activation of morphogenic factors like MLL, but also on the activity of methyltransferases as well as on general transcription factor (e.g., TFIIA) homeostasis [117,118].

A fourth putative asparaginase in humans is the 60-kDA lysophospholipase (official gene name: ASPG; hASNase1), which is similar to the *E. coli* EcASNase 1 and, therefore, belongs to the type 1 asparaginases. However, ASPG does not display the characteristic αββα-fold [119] and relies on a reaction mechanism other than Ntn-hydrolases; therefore, I does not belong to the superfamily [120].

To date, bacterial-derived asparaginases are prominent antineoplastic therapeutics in certain cancer treatments, as asparagine depletion from the serum hampers DNA and RNA synthesis of quickly dividing cancer cells and, subsequently, results in apoptosis [121]. However, *E. coli*-derived asparaginases often cause immunological complications in terms of neutralizing antibodies or even anaphylactic shock conditions, while the alternatively used *D. dadantii*-derived therapeutics exhibit shorter lifetime and lower efficacy [122]. Unfortunately, human candidates, which might replace the bacterial enzymes, have failed to comply with the kinetic properties (low *K*_M_ necessary for such a curative purpose so far, so further enzyme design is needed [123].

### 6.5. Family of γ-Glutamyl Transpeptidases (GGT)

The human genome encodes for a number of enzymatically active γ-glutamyl-transpeptidases (GTTs, Figure 2), as well as for several putatively inactive genes and three pseudogenes, respectively. Among the demonstrably active gene products, GGT1 is by far the best-characterized GGT protein and a prominent diagnostic serum marker, e.g., for liver damages, due to chronic hepatitis infections or long-term alcohol abuse. GGTs transfer γ-glutamyl groups (C5H8NO3; 4-amino-4-carboxybutanoyl) from glutathione, leukotrienes, or other donor substrates to acceptor substrates such as amino acids or short peptides, by forming a new isopeptide bond (transpeptidase reaction) or hydrolytically cleaving off the γ-glutamyl groups, indiscriminately, from all known γ-glutamyl substrates, thereby regulating the redox equilibrium in cells, inflammation events, and many other cellular (patho-)physiological processes [124,125]. By doing so, mouse Ggt1, which is expressed in various tissues, was shown to function mainly in glutathione homeostasis, while the other widely expressed Ggt5 is involved in leukotriene metabolism (e.g., LTC4/LTD4 turnover), as demonstrated in respective mouse knock out models [126,127]. Human patients, who lack GGT1 activity, are very rare and are described clinically by glutathionuria and LTD4 deficiency, as well as CNS symptoms such as seizures and mental retardation [128,129,130]. What kind of further substrates might be metabolized besides GGT1 and GGT5 remain to be elucidated, in regard to the further three GGT (GGT2, GGT6, GGT7) or GGT-like genes (GGTLC1-3) encoded in the human genome, of which the latter is encoded solely for a light chain (LC) variant [131]. Regardless of whether human, mouse, plant, or bacterial GGT, they all share the general αββα-Ntn-fold and also, unexceptionally, undergo autoproteolytic maturation into two subunits, which remain tightly associated. Likewise, all known GGTs own a conserved threonine residue (e.g., human GGT1 Thr381) acting as the nucleophile, which offers dual functionality. First, this particular threonine is needed for the autoproteolytic activation towards the mature amidase [132]. Consequently, the same threonine at the resulting N-terminus of the smaller subunit (20 kDa; Thr381–569) takes over the initial catalytic step by a nucleophilic attack of its side-chain hydroxyl oxygen on the γ-glutamyl amide bond, during the hydrolytic reaction cycle [124]. However, bacterial and eukaryotic GGTs differ besides post-translational modifications (glycosylation, disulfide bridges), with regard to their respective catalytic behavior, particularly in substrate-binding amino acids [132,133]. Such differences might be exploited for the development of novel GGT inhibitors, as human GGT1 was demonstrated to be a critical intensifier for diseases such as asthma, Parkinson’s disease, or cancer.

## 7. Concluding Remarks and Perspectives

Since the first report on Ntn-hydrolases as a new enzyme superfamily in 1995, a substantial number of already known (at that time), as well as enzymes identified later on, were assigned to that enzyme superfamily. All members of the Ntn-hydrolase superfamily are categorized due to their characteristic (Ntn)-fold, as well as their common way of activation and activity in terms of autoproteolytic processing, resulting in a nucleophilic amino acid at the newly emerged N-terminus [3,16]. Many human representatives of the Ntn-hydrolases are of particular medical interest—either for diagnostic purposes such as γ-glutamyl-transpeptidases, as disease-associated proteins such as the lysosomal AGA, or due to their (patho-)physiological impact, as can be seen in the different Ntn-subunits of the (immuno)-proteasome. However, we can expect a rising number of Ntn-family members in the near future, since, above all, the AI-based program AlphaFold 2 is on its way to revolutionize protein structure predictions [90] and, thus, will also help to identify novel Ntn-hydrolases—particularly on the basis of their characteristic Ntn-fold.

## Figures and Tables

**Figure 1 cells-11-01592-f001:**
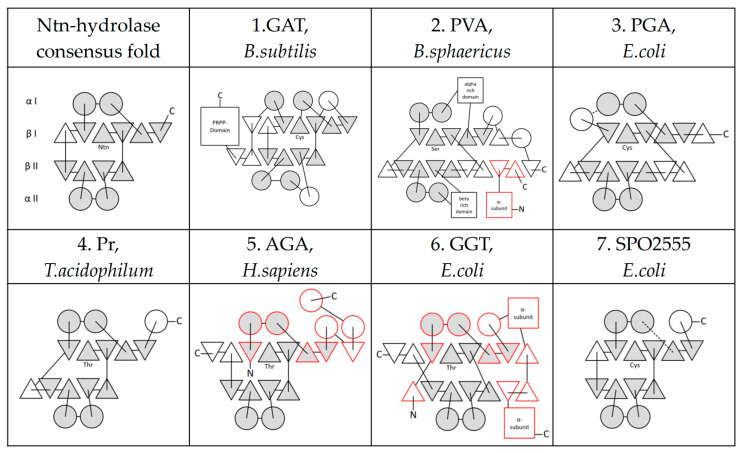
Secondary structural elements in Ntn-hydrolase subfamilies. The characteristic topology of αββα-secondary structure elements in N-terminal nucleophile (Ntn-)hydrolases consists of two α-helices (circles) in the αI-layer, five β-sheets (triangles) in the βI-layer, four β-sheets in the βII-layer, and two α-helices in the αII-layer (grey filling) [16]. Structurally, highly conserved elements between the various family members are filled in grey, while additional β-sheets (PVA) or α-helices (GAT) are white. The catalytic relevant Ntn residue is revealed at the newly generated N-terminus by autocatalytic cleavage in the βI-layer, either cleaving off only a few amino acids from the β-fragment (e.g., GAT, Pr) or forming a large α-subunit (red boxes) (e.g., AGA, GGT). The alpha subunit may even be a part of the highly conserved αββα-array. Dotted lines designate the presumed connections with diffuse electron density in the crystal structure. α-subunit elements are marked with a red frame, while β-subunit elements are marked with a black frame. Amino-termini (N), N-terminal nucleophiles (either Thr, Ser or Cys), and C-termini (C) are indicated.

**Figure 2 cells-11-01592-f002:**
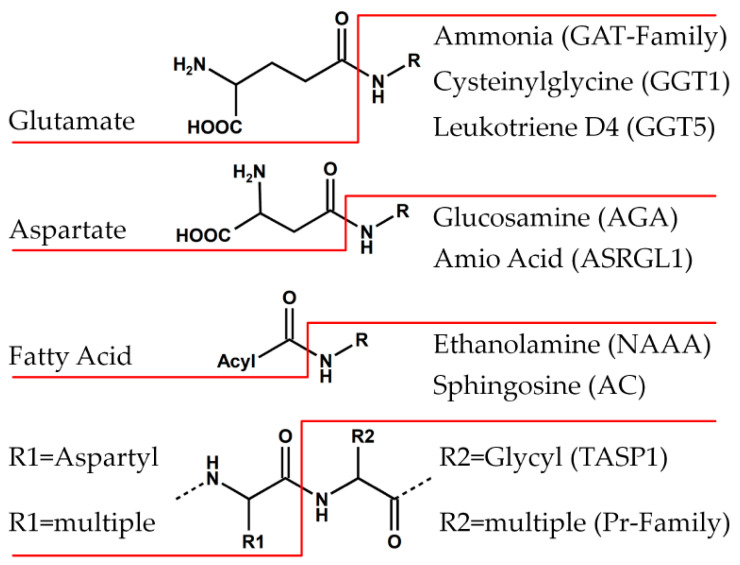
Cleaving products of human Ntn-hydrolases. Substrates of Ntn-hydrolases are cleaved at their amid bonds, releasing cleavage products. While the known carboxylic acid products are either amino acids (glutamate, aspartate, or peptides (Pr family)) or a fatty acid, the amine products show highly diversity (right side).

**Figure 3 cells-11-01592-f003:**
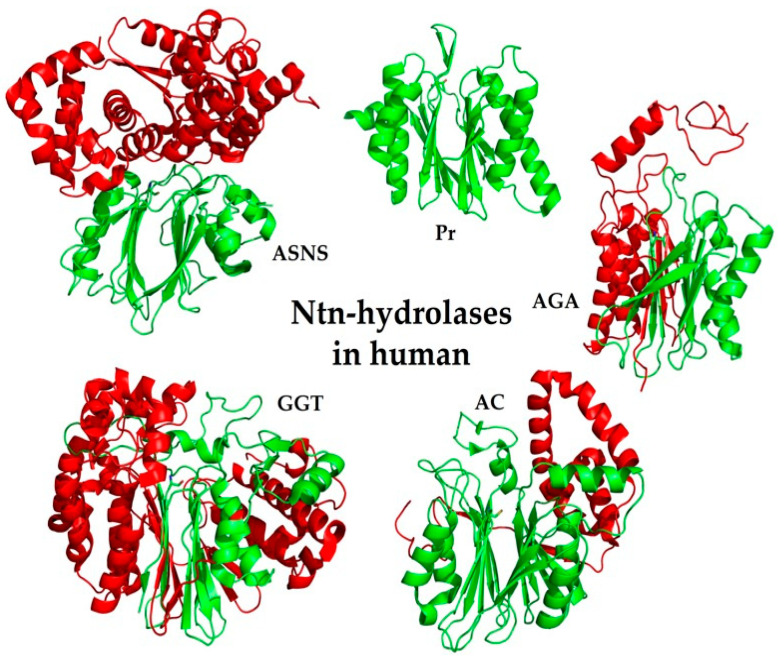
Representative 3-D structures of human Ntn-hydrolases. Three-dimensional structure of asparagine synthetase (ASNS, PDB 6GQ3), acid ceramidase (AC, PDB 5U7Z), proteome β-subunit (Pr, PDB 4r3O), lysosomal aspartyl glucosaminidase (AGA, PDB 1APY), and γ-glutamyl-transpeptidases (GGT1, PDB 4GDX). α-subunits are marked in red; β-subunits are marked in green.

**Figure 4 cells-11-01592-f004:**
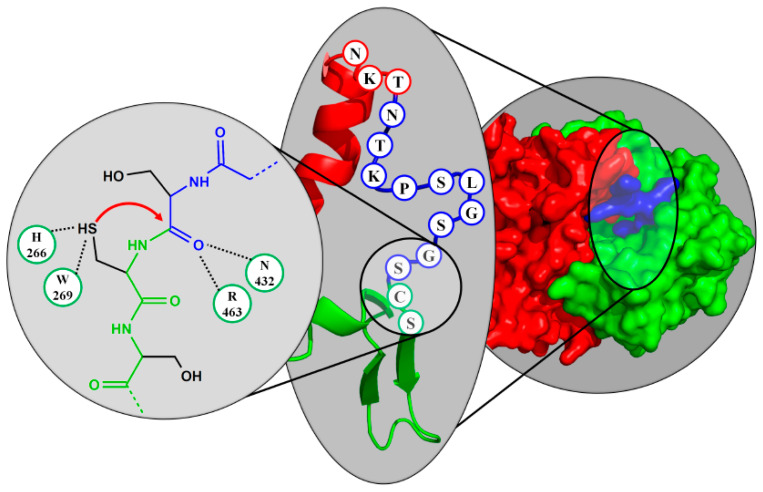
Sequence-related-function of Plbd2 active site and the disposable linker peptide. The crystal structure (predicted by alpha fold on the basis of UniProtKB Q3TCN2) represents the murine Plbd2 protein, consisting of the α-subunit marked in red, the β-subunit marked in green, and the disposable linker peptide marked in blue (background). The linker is sterically blocking the active pocket and, thus, needs to be detached from the precursor to activate the enzyme. The zoom into the linker region unveils the N-terminal nucleophile (C = cysteine 249), which is needed for autocatalytic cleavage of the linker peptide (foreground), according to PDB 3FGR. H266 and W269 stabilize the nucleophile; N432 and R463 form the oxyanion hole.

**Table 1 cells-11-01592-t001:** Ntn-hydrolase-like families with subfamilies according to SCOPe. The catalytic active N-terminal nucleophilic amino acid (catalytic residue) is indicated in the one letter code.

Ntn-Hydrolase-Like	Catalytic Residue	Reference Structure
Ntn-hydrolase Superfamily		
1. Class II glutamine amidotransferases	C	GAT, *B. subtilis* (PDB: 1GPH)
2. Penicillin V Acylases	C	PVA, *B. sphaericus* (PDB: 3PVA)
3. Penicillin G Acylases	S	PGA, *E. coli* (PDB: 1PNK)
4. Proteasome subunits	T	Pr, *T. acidophilum* (PDB: 1PMA)
5. (Glycosyl-)asparaginases	T	AGA, *H. sapiens* (PDB: 1APY)
6. Gamma-glutamyltranspeptidase-like	T	GGT, *E. coli* (PDB: 2DBU)
7. SPO2555	T	SPO2555, *S. pomerovi* (PDB: 2IMH)
Archaeal IMP cyclohydrolase		
1. Archaeal IMP cyclohydrolase PurO	N/A	MTH1020 *M. thermoautotrophicum* (PDB: 1KUU)
beta-Aminopeptidases		
1. DmpA like/BapA	S	DmpA, *O. anthropi* (PDB: 1B65)
2. Ornithin Acyltransferases/ArgJ	T	OAT, *E. coli* (PDB: IVZ6)

**Table 2 cells-11-01592-t002:** Human Ntn-hydrolases. Members of the human Ntn-hydrolase superfamily with Uniprot identifier and EC-number (if applicable), assigned to subfamilies according to MEROPS and SCOPe classifications. The known inactive proteasome subunits α1-α7, α4s, β3-β4, and β6-β7 are not listed. N/A = not applicable, tbd = to be determined; recommended protein names are stated first.

SCOPe Family	MEROPS Family	Gene	Protein Name	EC	UniProt
Class II glutamine amido-transferases (GAT) d.153.1.1	C44	ASNS	Glutamine-dependent asparagine synthetase (ASNS)	6.3.5.4	P08243
		ASNSD1	Asparagine synthetase domain-containing protein 1 (ASNSD1)	6.3.5.-	Q9NWL6
		PPAT	Glutamine phosphoribosyl-pyrophosphate amidotransferase (GPAT), Amidophosphoribosyltransferase PPAT, PUR1	2.4.2.14	Q06203
		GFPT1	Glutamine fructose-6-phosphate amidotransferase 1 (GFAT1) Gln-Fru6P-Transaminase 1 (GFPT1)	2.6.1.16	Q06210
		GFPT2	Glutamine fructose-6-phosphate amidotransferase 2 (GFAT2) Gln-Fru6P-Transaminase 2 (GFPT2)	2.6.1.16	O94808
Penicillin acylase (PVA) d.153.1.3	C89	ASAH1	Acid Ceramidase (AC, aCDase), N-acylsphingosine amidohydrolase 1 (ASAH1)	3.5.1.23	Q13510
		NAAA	N-acylethanolamine-hydrolyzing acid amidase (NAAA)	3.5.1.60	Q02083
	C95	PLBD1	Phospholipase B domain-containing protein 1, FLJ22662	tbd	Q6P4A8
		PLBD2	Phospholipase B domain-containing protein 2 (PLBD2), Phospholipase B-like 2 (PLBL2), P76, 66.3-kDa protein	tbd	Q8NHP8
	C69	SCRN1	Secernin-1, U34-dipeptidase homologue	N/A	Q12765
		SCRN2	Secernin-2, U34-dipeptidase homologue	tbd	Q96FV2
		SCRN3	Secernin-3, U34-dipeptidase homologue	tbd	Q0VDG4
Proteasome subunits (Pr) d.153.1.4	T1	PSMB6	Proteasome subunit β1, Y	3.4.25.1	P28072
		PSMB7	Proteasome subunit β2, Z	3.4.25.1	Q99436
		PSMB5	Proteasome subunit β5, X	3.4.25.1	P28074
		PSMB9	Proteasome subunit β1i, LMP2	3.4.25.1	P28065
		PSMB10	Proteasome subunit β2i, MECL-1	3.4.25.1	P40306
		PSMB8	Proteasome subunit β5i, LMP7	3.4.25.1	P28062
		PSMB11	Proteasome subunit β5t	3.4.25.1	A5LHX3
(Glycosyl) asparaginase (AGA) d.153.1.5	T2	AGA	Aspartylglucosaminidase (ASPG)	3.5.1.26	P20933
		ASRGL1	Isoaspartyl peptidase, Asparaginase-like protein 1, beta-aspartyl-peptidase, L-Asparaginase, ALP, hASNase3, CRASH, glial asparaginase	3.4.19.5	Q7L266
		TASP1	Aspartyl endopeptidase, threonine aspartase 1	3.4.25.-	Q9H6P5
Gamma-glutamyl-transpeptidase-like (GGT) d.153.1.6	T3	GGT1	Glutathione hydrolase 1 proenzyme, γ-Glutamyl transpeptidase 1	3.4.19.13	P19440
		GGT2	γ-Glutamyl transpeptidase 2	tbd	P36268
		GGT3	γ-Glutamyl transpeptidase 3	tbd	A6NGU5
		GGT5	γ-Glutamyl leukotrienase, γ-Glutamyl transpeptidase 5	3.4.19.14	P36269
		GGT6	γ-Glutamyl transpeptidase 6	tbd	Q6P531
		GGT7	γ-Glutamyl transpeptidase 7	tbd	Q9UJ14

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
