# Peer review of "The Human Ntn-Hydrolase Superfamily: Structure, Functions and Perspectives"

_cells, 2022, doi:10.3390/cells11101592_

Round 1
Reviewer 1 Report
The manuscript describe structure and mechanism of the different subfamily of N-terminal nucleophile (Ntn)-hydrolases that in humans are involved in the cleavage of amide bonds in a variety of biological relevant of bio-macromolecules.
Deregulation of the metabolic processes in which these enzymes are involved is at the root of several metabolic diseases. Consequently, these enzymes are of interest as potential therapeutic targets.
The manuscript is well written and of interest to readers of cells. In my opinion it can be published with minor revisions.
Literature should be revised
For example the references 1 and 2 do not seem relevant and should be removed or their content better explained in the text.
In other cases, numerous references are cited to support certain statements, but in some cases these references are very old.
For example, references 108-111 do not all appear to be all necessary, and some could be eliminated by reducing them to those that are indispensable or, if possible, using more recent references.
In some cases, metabolic steps whose deregulation is the basis of relevant pathologies should be illustrated with a figure or scheme to make the concepts described in the text clearer.
Author Response
Dear Reviewer,
thanks for your kind comments and suggestions.
We added two further figures in order to attract the readers attention.
The first figure (new Figure 1 shows the general substrates of the Ntn-hydrolases and also depicts the resulting products/ leaving groups.
The second new figure deals with the sequence-to-structure features of the Plbd2 active site and disposable linker peptide in order to demonstrate the autocleavage mechanism that results in the release of a short linker peptide and thus unveils the catalytic pocket.
We also edited the listed minor comments on spelling.
Best regards,
Torben Lübke
Reviewer 2 Report
The authors give a very extensive review on Ntn-hydrolase superfamily. Many general structural and functional details are given. The main part of the work deals with the five classes of enzymes which are found in human tissues. Here again, structure and function are in the focus, however also the connection to diseases is described.
The whole review is very well structured and supplies an enormous amount of information. Table 2 contains all relevant EC and accession numbers and can be used for further investigations.
Minor: Line 41 - delete one "By means"
Author Response
Dear Reviewer,
dear Sir or Madam,
thanks for your benevolent expert report. We changed you minor comments in the manuscript. However, due to the other two reviews we will add one or two figures in order to attract a wider readership.
Best regards
Torben Lübke
Reviewer 3 Report
Manuscript no.: cells-1685021
Title: The human Ntn-hydrolase superfamily: Structure, functions and perspectives
Article Type: Review Article
Dear Editor
In the present Review Manuscript, the authors collected and analyzed data, discussed the structure-related function of the Ntn-hydrolase superfamily, and gave perspective on finding a new family of these enzymes. The authors began with a discussion on the structure of Ntn-hydrolase, focusing on protein folding and catalytic acids on the active sites of the enzymes. Then, the authors discuss Ntn-hydrolase superfamily in humans. The discussion likely concentrates on the structure, enzymatic mechanism, and the effects of enzymes and their functions on human disease and disorder.
I think that this work is important in its field, and the authors give an insightful discussion. The manuscript is well-prepared. However, to attract the reader’s attention, I think that the authors might draw some figures such as sequence-to-structure and their classification and structure-related function and potential applications in medicine. Other minor comments are listed below. This work is useful for the reader.
- L 40: Remove “By means”, It is repetitive.
- L140-141: Recheck this sentence.
- L177: Change ‘if’ to “of”.
Author Response
Dear Reviewer,
thanks for your kind comments and suggestions.
We reduced the number of references as recommended but still left some "older" references as we also would like to give credit to some conceptual important original publications rather than to the newest reviews. We hope that we now managed to address both purposes by our reference selection. In total we reduced the absolute number by 15 references.
Regarding further figures (as also asked by another Reviewer).
We added two further figures in order to attract the readers attention. However, we did not addressed the pathophysiological mechanisms underlying the different diseases that are associated with the different human Ntn-hydrolases as these concepts are - from our point of view - too diverse to be portrayed in a reasonable and lucid figure. Moreover, we guess that such a figure would need additional text and would not suit to our general concept of the manuscript dealing with structure-related functions of the Ntn-hydrolase family.
Our newly inserted figure are of more general nature:
The first figure (new Figure 1 shows the general substrates of the Ntn-hydrolases and also depicts the resulting products/ leaving groups.
The second new figure (new Figure 3) deals with the sequence-to-structural features of the Plbd2 protein as an example of the active site and disposable linker peptide in order to demonstrate the autocleavage mechanism that results in the release of a short linker peptide and thus unveils the catalytic pocket.
Best regards,
Torben Lübke